# Construction and Validation of a Prognostic Gene-Based Model for Overall Survival Prediction in Hepatocellular Carcinoma Using an Integrated Statistical and Bioinformatic Approach

**DOI:** 10.3390/ijms22041632

**Published:** 2021-02-05

**Authors:** Eskezeia Yihunie Dessie, Siang-Jyun Tu, Hui-Shan Chiang, Jeffrey J.P. Tsai, Ya-Sian Chang, Jan-Gowth Chang, Ka-Lok Ng

**Affiliations:** 1Department of Bioinformatics and Medical Engineering, Asia University, No. 500, Lioufeng Rd., Wufeng, Taichung 41354, Taiwan; estu2003@gmail.com (E.Y.D.); jjptsai@gmail.com (J.J.P.T.); 2Department of Laboratory Medicine and Center for Precision Medicine, China Medical University and Hospital, No. 2, Yude Rd., North District, Taichung 404332, Taiwan; t34752@mail.cmuh.org.tw (S.-J.T.); d18448@mail.cmuh.org.tw (H.-S.C.); 3Department of Medical Research, China Medical University Hospital, China Medical University, No. 2, Yude Rd., North Dist., Taichung 404332, Taiwan; 4Center for Artificial Intelligence and Precision Medicine Research, Asia University, No. 500, Lioufeng Rd., Wufeng, Taichung 41354, Taiwan

**Keywords:** hepatocellular carcinoma, differential expressed gene, biomarker, prognosis, diagnosis, bioinformatics, survival analysis, risk model

## Abstract

Hepatocellular carcinoma (HCC) is one of the most common lethal cancers worldwide and is often related to late diagnosis and poor survival outcome. More evidence is demonstrating that gene-based prognostic models can be used to predict high-risk HCC patients. Therefore, our study aimed to construct a novel prognostic model for predicting the prognosis of HCC patients. We used multivariate Cox regression model with three hybrid penalties approach including least absolute shrinkage and selection operator (Lasso), adaptive lasso and elastic net algorithms for informative prognostic-related genes selection. Then, the best subset regression was used to identify the best prognostic gene signature. The prognostic gene-based risk score was constructed using the Cox coefficient of the prognostic gene signature. The model was evaluated by Kaplan–Meier (KM) and receiver operating characteristic curve (ROC) analyses. A novel four-gene signature associated with prognosis was identified and the risk score was constructed based on the four-gene signature. The risk score efficiently distinguished the patients into a high-risk group with poor prognosis. The time-dependent ROC analysis revealed that the risk model had a good performance with an area under the curve (AUC) of 0.780, 0.732, 0.733 in 1-, 2- and 3-year prognosis prediction in The Cancer Genome Atlas (TCGA) dataset. Moreover, the risk score revealed a high diagnostic performance to classify HCC from normal samples. The prognosis and diagnosis prediction performances of risk scores were verified in external validation datasets. Functional enrichment analysis of the four-gene signature and its co-expressed genes involved in the metabolic and cell cycle pathways was constructed. Overall, we developed a novel-gene-based prognostic model to predict high-risk HCC patients and we hope that our findings can provide promising insight to explore the role of the four-gene signature in HCC patients and aid risk classification.

## 1. Introduction

Hepatocellular carcinoma (HCC) is one of the most frequent malignant cancers and the fourth leading cause of cancer-related deaths worldwide [1]. There were more than 840,000 new cases of HCC and approximately 780,000 deaths annually [2]. Liver transplantation, tumor ablation, and surgical resection are currently the most effective treatment options to improve the survival of early-stage HCC. However, the majority of HCC patients are first diagnosed at late stage, and these patients experienced poor prognosis and high recurrence [3,4,5]. Therefore, HCC patients associated with poor prognosis need to be monitored and treated effectively to improve their prognosis.

Conventionally, risk factors including tumor-node-metastasis (TNM) staging, vascular invasion, and other parameters are commonly used for risk assessment of HCC patients [6]. However, these clinicopathological risk factors are not sufficient to classify between patients who have a high or low risk and fail to predict which patients are more likely to benefit from adjuvant chemotherapy. Therefore, in addition to clinicopathological risk factors, there is a strong demand to discover a novel and reliable signature to predict HCC patient prognosis and to identify the high-risk subgroup of HCC patients.

Prognostic models of gene expression have been constructed in many previous studies [7,8,9]. Li et al. built a prognostic model for patients with lung adenocarcinoma based on gene expression data [8]. L. Chen et al. obtained that the expression of the seven-gene model was associated with the prognosis of patients with clear cell renal carcinoma by lasso and Cox regression analysis [9]. For HCC, Chen et al. constructed a model to demonstrate that a nine-immune-gene-related signature can distinguish high- and low-risk groups [10]. A prognostic model based on a six-gene signature was used for overall survival prediction in HCC [6]. However, the development of multi-gene models to predict high-risk HCC patients are still unsatisfactory. Thus, it is essential to develop a comprehensive prognostic evaluation using a variety of prognostic methods to identify more potentially informative genes.

Recently, the development of next-generation sequencing and microarray technologies have assisted researchers in exploring genetic alterations in tumorigenesis and identifying novel biomarkers for several diseases [10,11]. Meanwhile, Cox proportional hazard model (CPHM) is the most broadly used method in survival analysis. However, when we deal with high-dimension data such as genomic data, CPHM is not the most appropriate method to select prognostic genes because of overfitting problems [11,12]. A number of penalized methods such as lasso [12], adaptive lasso [13] and elastic net [14] can eliminate this shortcoming and a multivariate Cox regression with three hybrid penalties—elastic net, lasso and adaptive lasso methods—were applied in our analysis.

In this study, we aimed to identify and validate differentially expressed genes (DEGs) associated with overall survival (OS) based on genome-wide expression data of HCC patients. We initially identified consistently DEGs based on multiple cohorts of the gene expression omnibus (GEO) and The Cancer Genome Atlas (TCGA) datasets. We used a univariate Cox regression analysis, multivariate Cox regression model with three hybrid penalties including elastic net, lasso and adaptive lasso algorithms as well as best subset regression (BSR) to screen a multiple gene signatures with the smallest Akaike information criterion (AIC) values. The risk score was constructed through a linear combination of the gene expression level and the multivariate Cox regression coefficient. The risk predictive model was validated in various aspects using internal and external datasets. Finally, functional enrichment analysis was conducted to elucidate the biological pathways of the identified novel four-gene prognostic signature and the co-expressed DEGs. We hope that this risk predictive model can help to identify HCC patients with a higher risk of mortality and delivers more insights into the development and progression this disease.

## 2. Results

### 2.1. Identification and Validation of Potential DEGs

The overall workflow of this study is shown in Figure 1. Initially, a total of four different gene expression datasets of HCC patients including GSE112790, GSE84402 and GSE45267 and TCGA-HCC were collected. After GEO and TCGA data filtering, quality assessment and normalization, differential gene expression analysis was performed using limma R package. The distribution of the DEGs in each dataset is shown (Figure 2a–d) and a total of 711 DEGs in GSE112790, 696 DEGs in GSE45267, 1361 DEGs in GSE84402, and 1928 DEGs in TCGA-HCC were identified as compared to normal tissue samples. Subsequently, from the intersection analysis, we found 339 common DEGs in HCC, satisfying the criteria of absolute value of log2FoldChange (LFC) > 1.5 and false discovery rate (FDR) < 0.01 (Figure 2e and Appendix A).

### 2.2. Identification of Prognosis-Related DEGs

According to prognostic gene selection methods discussed in the Materials and Methods section, we identified a total of 75 prognosis-associated DEGs out of 339 DEGs satisfying the criteria of hazard ratio (HR) > 1 or HR < 1 and *p*-value < 0.05 based on univariate Cox regression analysis in TCGA dataset of HCC patients (*n* = 359). To further identify informative prognostic genes associated with the prognosis of HCC patients, three popular feature selection algorithms including elastic net, lasso, and adaptive lasso with 10-fold cross-validation were implemented to identify the optimal λ values ($\lambda_{\mathrm{opt}}^{\mathrm{Enet}}=0.070,{\;\lambda }_{\mathrm{opt}}^{\mathrm{Lasso}}=0.039$ and $\lambda_{\mathrm{opt}}^{\mathrm{Adlasso}}=0.00432$) that derived from minimum mean cross-validation errors, which were associated with 13 DEGs, 9 DEGs and 6 DEGs that significantly correlated with OS, respectively (Table 1; Figure 3a–c). The union of OS-related genes predicted by the three algorithms (including *FAM83D*, *CDC20*, *TPX2*, *UBE2S*, *LECT2*, *ANXA10*, *DNASE1L3*, *PON1*, *CD5L*, *CYP2C9*, *ADH4*, *CFHR3*, *GHR*, and *LCAT*) were used for best subset regression (BSR) analysis. Then, BSR analysis was performed to identify a four-gene prognostic model with the minimum Akaike information criterion (AIC) value, namely Family with sequence similarity 83-member D (*FAM83D*), Alcohol dehydrogenase 4 (*ADH4*), Alcohol dehydrogenase 4 (*ADH4*), Growth hormone receptor (*GHR*) and Lecithin-cholesterol acyltransferase (*LCAT*).

The median of the four-gene signature expression value was considered as a cutoff to stratify the TCGA HCC samples into high and low expression groups and then survival analysis was performed to assess the survival difference between these two groups. The over-expression of *FAM83D* (HR = 1.7, *p* = 0.0046) was correlated with unfavorable prognosis of patients with HCC (Figure 3d) whereas high expression levels of *LCAT* (HR = 0.49, *p* = 0.0001), *GHR* (HR = 0.68, *p*-value = 0.033), and *ADH4* (HR = 0.49, *p*-value = 0.0001) were correlated with better prognosis of HCC patients (Figure 3d–g).

### 2.3. Genetic Alterations and Survival-Related Gene Expression Profiles

The comparison of the four-gene expression level between HCC and normal tissue shown in Table 2 and Appendix A demonstrate that *FAM83D* was significantly over-expressed in tumors while *LCAT, GHR* and *ADH4* were significantly downregulated in multiple GEO and TCGA datasets when compared with normal tissue (*p* < 0.001). Besides, we utilized the cBioportal database (https://www.cbioportal.org/, accessed on 1 January 2021) to examine genetic alterations of the identified four-gene signature across multiple datasets (Figure 4a). Amplification and missense mutations were observed in *LCAT*, *ADH4*, *GHR*, and *FAM83D*, while truncating mutation was commonly observed in *LCAT*, *GHR*, and *FAM83D*. Additionally, deep deletion was observed in *ADH4.*

### 2.4. Development and Estimation of the Four-Gene Signature

We performed multivariate Cox regression analysis on the identified four-gene signatures to evaluate whether each gene could reveal a significant prognosis prediction relevance for HCC patients and then we developed a risk score (RS) model using the four-gene expression profiles and their Cox regression coefficients as: RS = 0.2277∗EXP(*FM83D*) − 0.1554∗EXP(*LCAT*) − 0.0584∗EXP(*ADHA*) − 0.1137∗EXP(*GHR*), where the constant denotes the gene coefficient obtained from multivariate Cox regression and EXP denotes gene the expression level. The RS of every patient was calculated and the patients were classified into the high-risk group (*n* = 180) and low-risk group (*n* = 179). Then survival analysis indicated that the higher-risk group had poor overall survival when compared with lower-risk patients (*p* < 0.0001; Figure 4b). Additionally, we performed risk stratification in HCC patients with TNM stage and tumor grade, and performed KM survival analysis. The high-risk group had poor prognosis compared with the low-risk group in stage I and II (*p*-value < 0.05), stage III and IV (*p*-value < 0.05), grade 1 and 2 (*p*-value <0.05), and grade 3 and 4 (*p*-value < 0.05) (Appendix A). Time-dependent ROC curve analysis of the four-gene prognostic signature for 1-year survival showed that the AUC value of the four-gene prognostic signature was higher than that of a single-gene signature (Appendix A). Furthermore, the ROC analysis showed that the four-gene signature AUC value of the time-dependent ROC curve was 0.784, 0.732 and 0.733 for 1-year, 2-year and 3-year survival, respectively (Figure 4c). Besides, the diagnostic performance of the four-gene signature was evaluated by ROC curve analysis and the result revealed that the four-gene signature had higher diagnostic prediction performance when compared with a single gene, which demonstrates that the multi-gene signature had better diagnostic performance to classify HCC from normal tissue (Figure 5a). Finally, univariate and multivariate analysis showed that the four-gene signature can be used as independent risk factor (Table 3). The overall result demonstrated that the risk prediction based on a four-gene signature can be used for risk assessment for HCC patients.

### 2.5. External Prognostic and Diagnostic Validation of Four-Gene Signature

To validate the prognostic and diagnostic prediction values of the four-gene signature, we utilized the International Cancer Genome Consortium (ICGC) dataset as an external validation set. The expressions levels of the four genes between tumor and normal tissue samples were compared (Figure 5b), which were consistent the results of TCGA and GEO datasets. Then, the risk model was developed using this dataset and the risk score of each patient was calculated. The KM survival analysis showed that high-risk patients had poor OS rate when compared with the low-risk subgroup (*p*-value < 0.001; Figure 5c). The ROC analysis showed that the four-gene signature AUC value of the time-dependent ROC curve was 0.634, 0.674 and 0.671 for 1-year, 2-year and 3-year survival, respectively (Figure 5d). Furthermore, the ROC curve analysis showed the four-gene signature AUC value of ROC curve was 0.952, which indicates that the four-gene signature yielded a stronger diagnostic performance to classify HCC from normal tissue (Appendix A).

### 2.6. External Prognostic and Diagnostic Verification of the Four-Gene Signature

To verify the four-gene signature, we used 158 HCC and 12 normal samples of patients as external verification dataset from China Medical University Hospital (CMUH). We first examined gene differential expression between HCC and normal tissue samples and the results were consistent with the aforementioned findings (Figure 6a). Besides, the four-gene risk model was used to calculate risk score of each patient and then patients were classified into high-risk and low-risk subgroups. The KM survival analysis showed that the high-risk group had poor prognosis relative to the low-risk group (*p*-value < 0.001; Figure 6b). The ROC analysis showed that the four-gene signature AUC value of the time-dependent ROC curve was 0.865, 0.854 and 0.779 for 1-year, 2-year and 3-year survival, respectively (Figure 6c). Moreover, the AUC value of the four-gene signature was 0.985, which revealed that the four-gene signature had strong diagnostic performance to classify HCC tissue from normal tissue (Figure 6d). Overall, the results demonstrate that the four-gene signature can be used for predicting the prognosis and diagnosis of HCC, which indicate that the four-gene signature may serve as a potential biomarker in HCC patients.

### 2.7. Identification of Biological Pathways of Four-Gene Signture and Thier Coe-Expressed Degs

To understand the functional roles of the identified four-gene signature and the co-expressed genes, we first performed correlation analysis between the four-gene signature and the remaining DEGs in HCC. We obtained many genes, which were co-expressed with four prognostic genes (Pearson’s correlation coefficient, r ≥ 0.5 and *p*-values < 0.05) (Appendix A). Next, the four-gene signature and its co-expressed genes were used for enrichment analysis. The gene ontology (GO) analysis of the four-gene signature and the co-expressed genes were mainly involved in cell division, fatty acid beta-oxidation using acyl-CoA dehydrogenase, mitotic nuclear division, DNA replication, and other key biological processes were used for the GO analysis (Appendix A). Meanwhile, Kyoto Encyclopedia of Genes and Genomes (KEGG) pathway analysis of the four-gene signature and the co-expressed genes was mainly enriched in retinol metabolism, metabolic pathways, complement and coagulation cascades, cell cycle and chemical carcinogenesis, and other biological pathways (Table 4). We also performed gene set enrichment analysis (GSEA) between high-risk and low-risk HCC subgroups. The high-risk subgroup with low expression levels of LCAT, ADH4, and GHR was enriched in the organic hydroxyl compound metabolic process, lipid metabolic process, alcohol metabolic process, and cellular hormone metabolic process (Figure 7a–d), while FAM83D upregulation in the high-risk group involved in cell cycle, cell cycle regulation, and cytoskeleton organization (Figure 7f–h).

## 3. Discussion

HCC remains the most common malignant cancer worldwide. Traditional risk factors such as TNM staging and vascular invasion used to predict prognosis of HCC patients. However, as discussed above, pathological risk factors alone are not sufficient to predict prognosis. Identifying robust prognostic biomarkers and the construction of an effective prognostic model to predict the prognosis of HCC is urgently needed for clinical practice.

In this study, we first identified robust DEGs from multiple cohorts of GEO and TCGA by reducing noise arising from sequencing platform type, data selection and data normalization. Prognosis-related DEGs screened from the univariate Cox regression method in TCGA dataset from the identified DEGs and then multivariate Cox regression with elastic net, lasso and adaptive lasso penalties were applied with 10-fold cross-validation to identify 12 informative survival-related DEGs including *FAM83D*, *CDC20*, *TPX2*, *UBE2S*, *LECT2*, *ANXA10*, *DNASE1L3*, *PON1*, *CD5L*, *CYP2C9*, *ADH4*, *CFHR3*, *GHR* and *LCAT*. Finally, BSR analysis screened out the final optimal novel four-gene signature (*FAM83D*, *ADH4*, *GHR* and *LCAT*), where the procedure and feature section methods were novel as compared with most previous research, and two verification analyses were performed using external datasets, to show the reproducibility of the results.

To the best of our knowledge, there has not been any study using multivariate Cox regression with elastic net, lasso and adaptive lasso penalties screening methods like ours to identify informative DEGs associated with prognosis of HCC patients. The four novel genes are significantly associated with prognosis of HCC patients. While FAM83D is a risky prognostic gene, *ADH4*, *GHR* and *LCAT* are protective prognostic genes. The biological roles of the four identified genes have been reported in the existing literature. The *FAM83D* gene has been confirmed as a target in various cancers including HCC [15,16]. *FAM83D* is overexpressed and related to gender, TNM stage, tumor recurrence and prognosis in HCC [17]. Overexpressed of *FAM83D* leads to amplified cell proliferation, migration and metastasis of ovarian cancer cells and high expression correlated with tumor stage and grade [18]. Downregulated *LCAT* is strongly related to poor survival in HCC [19]. *ADH4* is significantly downregulated in HCC when compared with non-cancerous tissue, besides the *ADH4* protein expression is lower in HCC [20]. Wei et al. reported that lower expression of *ADH4* was associated with poor prognosis in HCC [21]. Furthermore, the level of *GHR* is upregulated in breast cancer [22,23] and lung cancer [24]. The correlation analysis of four prognostic genes and DEGs in HCC showed that four prognostic genes significantly correlated with several DEGs in HCC, which indicated that the identified novel four-gene signature may be the driver of many genes during progression and development of HCC. Our functional enrichment analysis of the identified four-gene prognostic signature and the co-expressed DEGs demonstrated that the novel four-gene prognostic signature and/or the co-expressed genes were mainly enriched in the metabolic and cell cycle pathways, which may suggest that the disturbance of four novel genes may promote HCC formation and development through affecting metabolic and cell cycle pathways.

Compared to previous research, our study used a different approach to identify novel gene signatures [25,26]. Our prognostic model building strategy include multivariate Cox regression with elastic net, lasso and adaptive lasso penalties can identify robust prognostic genes by reducing the multicollinearity problems in the genome [27], where multicollinearity is a situation in which two or more prognostic genes are pairwise correlated. The BSR analysis identified an optimal four-gene signature predictive model, which had the minimum AIC values. After pinpointing the four prognostic genes, a four-gene prognostic model was constructed and examined for its prognostic value in HCC patients. The high-risk groups of patients had significantly worse prognosis than the low-risk group of patients. Additionally, the prediction of the four-gene-based risk model could be utilized in the stratified HCC patients such as stage I and II, stage III and IV, grade 1 and 2 and grade 3 and 4. We observed that high-risk patients were associated with poor prognosis compared with low-risk patients in stage I and II, stage III and IV, grade 1 and 2 and grade 3 and 4, which demonstrates that the novel four-gene-based risk model could be used to classify HCC patients into different risk groups in these subgroups. The AUC value of the risk prediction performance of the four-gene signature for 1-year, 2-year and 3-year survival revealed good prediction performance. Furthermore, the four-prognostic gene signature showed high diagnostic performance to classify HCC from normal samples. The univariate and multivariate Cox regression analyses indicated that the four-gene signature could be an independent risk factor to evaluate the prognosis. Besides, we used external validation datasets from ICGC and CMUH to verify the prediction performance of the four-gene signature. However, we acknowledge some limitations in our study. Frist, PCR-based functional experiments should be conducted to reveal the role of the four novel genes in cancers formation and development. Second, we did not include treatment effect while developing the risk predictive model due to a lack of complete medical records.

## 4. Materials and Methods

### 4.1. HCC Sample Source

In this study, the gene expression profiles of HCC samples were retrieved from TCGA (https://gdac.broadinstitute.org/, accessed on 1 January 2021), GEO (https://www.ncbi.nlm.nih.gov/geo/, accessed on 1 January 2021), and ICGC (https://icgc.org/, accessed on 1 January 2021) databases. We selected three gene expression profiles of HCC patients from GEO including GSE112790 [28], GSE84402 [29], and GSE45267 [30,31]. The inclusion criteria of three datasets were as follows: the human samples were classified into the HCC group and adjacent or nontumor groups; samples sizes above fourteen for HCC and nontumor groups were included in each dataset and mRNA expression profiling was used to examine each the samples. The gene expression profiles of TCGA-HCC dataset consist of 371 HCC and 50 normal tissue samples. The gene expression data of three GEO datasets and TCGA were used to identify DEGs. The normalized gene expression data and the clinical medical records (age, gender, pathological stage, OS time and status) of TCGA-HCC dataset (*n* = 359) with complete clinical information was used as a training set to develop prognostic models. The gene expression profiles of ICGC LIRI-JP dataset consist of 243 HCC and 202 normal tissue samples. The normalized gene expression data and the clinical information (including age, gender, pathological stage, OS time and status information) of ICGC LIRI-JP dataset (*n* = 232) with complete clinical information was used to validate prognostic models. The detailed description of the datasets used our study is shown in Appendix A.

### 4.2. Data Preprocessing and Identfication of DEGs

The datasets were pre-processed separately depending on the profiling methods for different platforms. The robust multi-array average technique was used for the standardization of GEO-Affymetrix platform-based gene expression profiles. We used logarithm base two transformation for the normalized RNA-Seq by Expectation–Maximization (RSEM) values of the genes of TCGA data. We removed minimally expressed genes if they had zero reads in more than 20% of the samples in each dataset. Then, normalized gene expression profiles of the three GEO and TCGA datasets of HCC sample of patients were used to screen DEGs based on the “limma” package (in R platform) [32]. Gene differential analysis between HCC tissue samples and normal tissue samples were compared and genes with absolute log fold change (LFC) values ≥1.5 and adjusted *p*-values < 0.01 were considered as DEGs.

### 4.3. Prognosis-Related Gene Selection and Development of Prognostic Model

Feature (gene) selection is a key process to improve prognosis prediction performance by avoiding noise or irrelevant features. The normalized RSEM values of the genes were further transformed using log2RSEM for subsequent survival analysis. Then, normalized gene expression data were utilized to estimate the association between gene expression level and the survival time of HCC patients based on TCGA dataset (*n* = 359). A univariate Cox proportional hazard model was employed to identify the prognosis-associated DEGs satisfying the criteria of hazard ratio (HR) > 1 or HR < 1 and *p*-value < 0.05. Finally, the prognosis-related DEGs were used in subsequent analysis.

After identifying the candidate survival-related genes, we proposed multivariate Cox regression model with lasso, adaptive lasso and elastic net penalties to select the most significant survival-related genes in TCGA dataset for prognostic model construction. Three methods are penalized approaches that are used to avoid the overfitting problem [12,13,14,33]. Lasso and adaptive lasso are used to select prognostic genes through the shrinkage of some of the irrelevant genes’ regression coefficients to zero. Adaptive lasso imposes a greater penalty in comparison with the lasso algorithm, which further reduces less relevant prognostic genes in such a way that the resulting coefficient estimates are sparse. Though the lasso algorithm is widely used to select prognostic-associated genes, it has a weakness in selecting prognostic genes when there is a multicollinearity among genes in the genome. In other words, a number of genes in a genome are pairwise correlated due to the complex interaction and this situation creates the problem of multicollinearity. In contrast, the elastic net algorithm is a hybridized version of the ridged [34] and lasso algorithms [13], which was proposed to identify informative genes when there is a problem of multicollinearity in the genomic data [27]. Since each algorithm has the strength to select informative genes with vigorous predictive power, a multivariate Cox proportional hazard regression model with lasso, adaptive lasso and elastic net penalties was proposed to construct a multi-gene signature for predicting prognosis using “glmnet” package of R software [35]. To increase the robustness of the results, 10-fold cross validation was implemented to estimate the optimal lambda value(λ) for each method that came from the minimum mean-squared prediction error.

Then, the union of prognostic-related genes selected by three methods with 10-fold cross-validation was further analyzed using best subset regression (BSR) algorithm. The BSR algorithm can compare all possible generated prognostic models based upon the identified prognosis-associate genes. Suppose we obtained a total of K prognostic-associated genes from lasso, adaptive lasso and elastic net algorithms. The BSR method identified an optimal prognostic model based on the following procedure.

Consider τ = 1, 2, 3; τ = K.

Construct all possible models C (K, τ) using τ subset of identified genes.Calculate Akaike information criterion (AIC) for each constructed model in a.Choose an optimal prognostic model, whose AIC is the smallest τ prognostic genes using the “glmulti” package (in R platform) [36].

### 4.4. Construction and Estimation of Prognostic Gene Signature

TCGA dataset (*n* = 359) and external validation set (*n* = 232) were used to confirm the predictive ability of multi-gene signature in HCC. All regression coefficients were obtained from optimal multivariate Cox regression model using TCGA dataset and validation datasets. Then, the risk score (RS) for each HCC patient was calculated using the formula: RS = $\underset{i}{\sum }X_{i}\beta_{i}$, where $\beta_{i}$ is the regression coefficient of the ith gene and $X_{i}$ is the log2-transformed expression value of the ith prognostic gene. The HCC patients were then stratified into high-risk and low-risk subgroups, based on the median RS as a cutoff. Kaplan–Meier (KM) survival analysis was used to evaluate the survival difference between the high-risk and low-risk subgroups. The prediction performance of the risk prediction model was evaluated by the area under the time-dependent receiver operating characteristic (AUROC) curve. The multivariate Cox analysis adjusting clinicopathological variables based on Equation (1) was used to confirm whether RS could be used as an independent prognostic risk factor in HCC patients.
(1)$$h\left(t,x^{\prime }\beta \right)={\;h}_{0}\left(t\right)\exp\left(\beta^{\prime }x\right)$$
(2)$$x^{\prime }\beta =\beta_{\mathrm{Stage}}\mathrm{Stage}+{\;\beta }_{T\_\mathrm{stage}}T\_\mathrm{stage}+\beta_{\mathrm{grade}}\;\mathrm{grade}+\beta_{\mathrm{RS}}\;\mathrm{RS}$$
where $h\left(t,x^{\prime }\beta \right)$ is the multivariate Cox model associated with covariates at time t (stage, tumor stage, neoplasm grade, and RS), and β denotes the regression coefficient of stage, tumor stage, tumor grade, and RS.

### 4.5. Evaluation of the Diagnostic Performance of the Multi-Gene Signature in HCC

To evaluate the diagnostic capacity of the multi-gene signature, we first computed a logistic regression model to extract the coefficients of the identified prognostic genes based on TCGA dataset of HCC patients including both HCC and normal tissue samples (*n* = 421) and ICGC LIRI-JP dataset including both HCC and normal tissue samples as external validation set (*n* = 445); then, the RS model was derived to evaluate the diagnostic performance in classifying HCC tissue from normal tissue.

### 4.6. Clinical HCC Patient Samples

To verify the identified optimal gene expression levels, we analyzed a total of 158 HCC and 12 normal tissue samples from CMUH, Taiwan. The prognostic model was verified based on the CMUH-HCC dataset (*n* = 80) with complete survival time (at least more than 30 days follow-up). We also verified the diagnostic performance of the risk mode using 158 HCC and 12 normal tissue samples. All the patients who participated in this study signed informed consent and all the protocols for the trial was approved by the Human Research Ethics Committee of CMUH (CMUH106-REC1-053, 6 May 2020). The tumor and adjacent normal tissues were then collected from patients who underwent liver surgical resection. All tissues were then obtained immediately after surgical resection and frozen at −80 °C until RNA extraction.

### 4.7. RNA Sequencing (RNA-Seq) and Data Analysis

For CMUH dataset, RNA was extracted from clinical tissue samples using the NucleoSpin^®^ RNA Kit (MACHEREY-NAGEL GmgH, Düren, Germany) following the manufacturer’s instructions. The quality, quantity, and integrity of the total RNA were evaluated using the NanoDrop1000 spectrophotometer and Bioanalyzer 2100 (Agilent Technologies, Palo Alto, CA, USA). Samples with an RNA integrity number >6.0 were used for RNA-seq. An mRNA-focused, barcoded library was generated using the TruSeq strand mRNA Library Preparation Kit (Illumina). The libraries were sequenced on the Illumina Nova Seq 6000 instrument (Illumina), using 2x151 bp paired-end sequencing flow cells following the manufacturer’s instructions. After obtaining the RNA fastq data, we performed quality control via Trimmomatic v0.38 [37] with the following parameters: LEADING:3, TRAILING:3, SLIDINGWINDOW:4:15, MAXINFO:40:0.2, MINLEN:100, AVGQUAL:20. The passed reads were then aligned with the Ensembl 84 release GRCh38 reference genome by HISAT2 [38] and the gene expression profiles were qualitied by StringTie [39] with Homo_sapiens.GRCh38.95.chr_patch_hapl_scaff.gff3. Finally, transcripts per million (TPM) was applied for normalizing the gene expression and then further transformed using logarithm base two.

### 4.8. Biological Characterization of the Identified Prognostic Genes

To elucidate the biological roles of the identified prognostic DEGs and their co-expressed DEGs, pathway analysis using the Database for Annotation, Visualization, and Integrated Discovery (DAVID) [40] was performed. Gene Set Enrichment Analysis (GSEA) V4.1.0, https://www.gsea-msigdb.org/gsea/index.jsp, accessed on 1 January 2021) was used to understand whether the expression levels of the predefined genes showed a significant difference between the high-risk and low-risk subgroups of HCC patients. The gene set “C5.bp.v7.1symbols.gmt” obtained from the molecular signature database (https://www.gsea-msigdb.org/gsea/msigdb/index.jsp, accessed on 1 January 2021) was chosen for enrichment analysis. A *p*-value < 0.05 was used to select significant enrichment results.

### 4.9. Statistical Analysis

Data preprocessing and normalization, and statistical analyses were performed using R software. Welch’s *t*-test was used for differential gene analysis. KM survival analysis with a log rank test was used to evaluate the survival of the HCC patient subgroups. Pearson’s correlation coefficient (r) was used to identify genes that were co-expressed with identified prognostic gene signatures. A *p*-value < 0.05 was used to select significant results, unless otherwise stated.

## 5. Conclusions

In summary, we identified and validated four novel prognostic gene signature using a novel integrated statistical methodology approach. Our findings demonstrate that the four-gene prognostic signature is a potential biomarker for patients with HCC. The four-gene-based risk model is effective to stratify patients with HCC into high- and low-risk groups. Therefore, this risk predictive model could potentially help in clinical practice for predicting the survival of individual patients and facilitating the development of better treatment options for HCC patients.

## Figures and Tables

**Figure 1 ijms-22-01632-f001:**
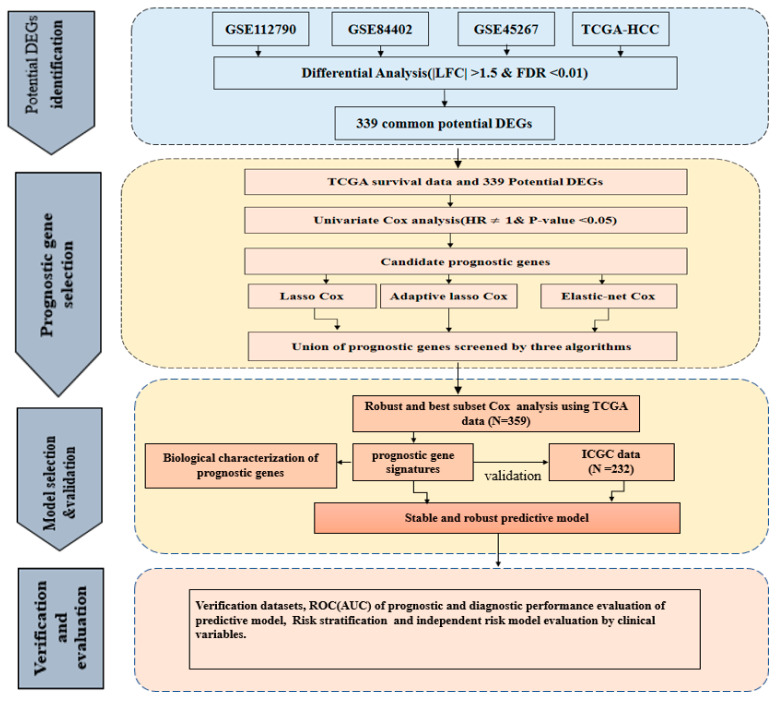
The workflow of our study.

**Figure 2 ijms-22-01632-f002:**
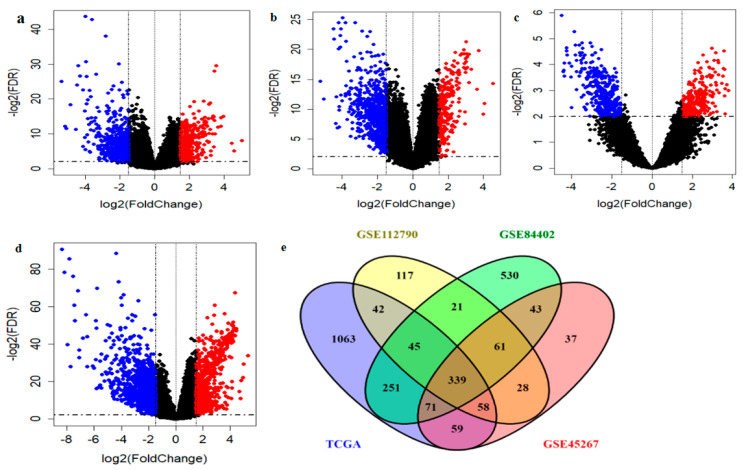
DEGs analyses in three gene expression omnibus (GEO) and The Cancer Genome Atlas (TCGA) datasets. DEGs were screened based on criteria |log2FoldChange| > 1.5 and FDR < 0.01. (**a**–**d**) The volcano plots of DEGs distribution are displayed in GSE112790 (**a**), GSE45267 (**b**), GSE84402 (**c**) and TCGA-HCC (**d**). Common DEGs from four datasets (**e**). Red dots indicate genes with FDR < 0.01 and log2FoldChange > 1.5; Blue dots indicate genes with FDR < 0.01 and log2FoldChange < −1.5; FDR, false discovery rate; DEGs, differentially expressed genes.

**Figure 3 ijms-22-01632-f003:**
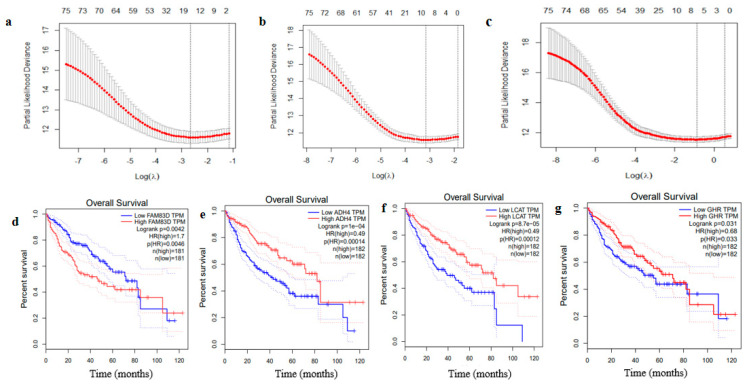
Identifying relevant prognostic genes from hepatocellular carcinoma (HCC) patients based on TCGA dataset. Tenfold cross validation (10-CV) gained prognostic genes by minimum lambda values for elastic net (**a**), lasso (**b**), adaptive lasso (**c**). X-axis shows that regularization process and y-axis show partial likelihood deviance. The minimum value numbers of prognostic genes by three algorithms are shown with two vertical lines that show one standard deviation from the minimum values. Kaplan–Meier (KM) survival analysis of high expression and low expression identified by BSR model (**d**–**g**). BSR, Best subset regression.

**Figure 4 ijms-22-01632-f004:**
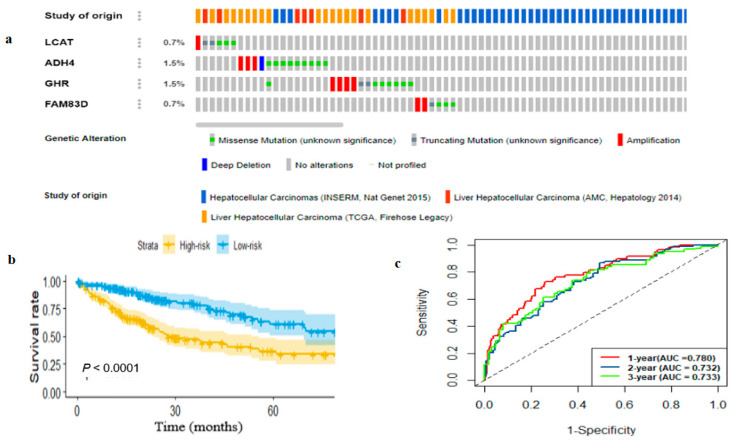
Prognostic assessment of four-gene signature for HCC. The four-gene signature genetic alteration in multiple HCC studies identified by cBioportal (**a**). Kaplan–Meier (KM) survival analysis for high-risk and low-risk subgroups stratified by RS model in TCGA dataset (**b**). Validation of four-gene signature by time-dependent receiver operating characteristic curve (ROC) analysis in TCGA dataset (**c**). RS, risk score.

**Figure 5 ijms-22-01632-f005:**
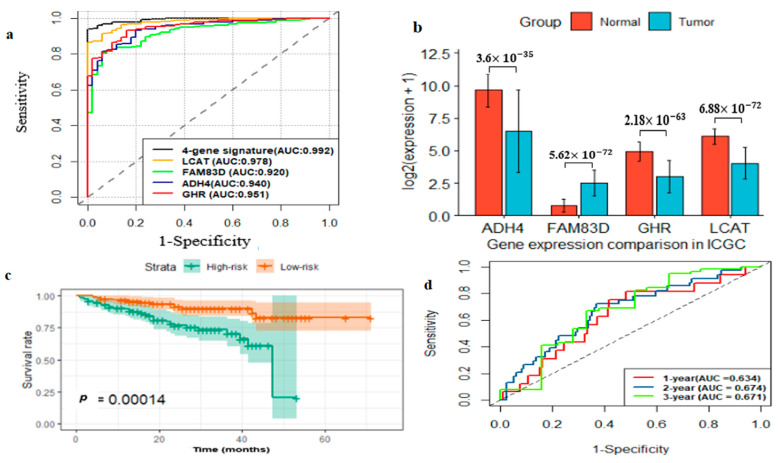
Prognostic and diagnostic assessment of four-gene signature for HCC. Four-gene signature diagnostic performance in classifying HCC and normal tissue in TCGA dataset (**a**). Validation of four-gene signature expression between HCC and normal tissue using external International Cancer Genome Consortium (ICGC) dataset (**b**). Kaplan–Meier (KM) survival analysis for high-risk and low-risk subgroups stratified RS model in ICGC dataset (**c**). Validation of four-gene signature by time-dependent ROC analysis in ICGC dataset (**d**). RS, risk score.

**Figure 6 ijms-22-01632-f006:**
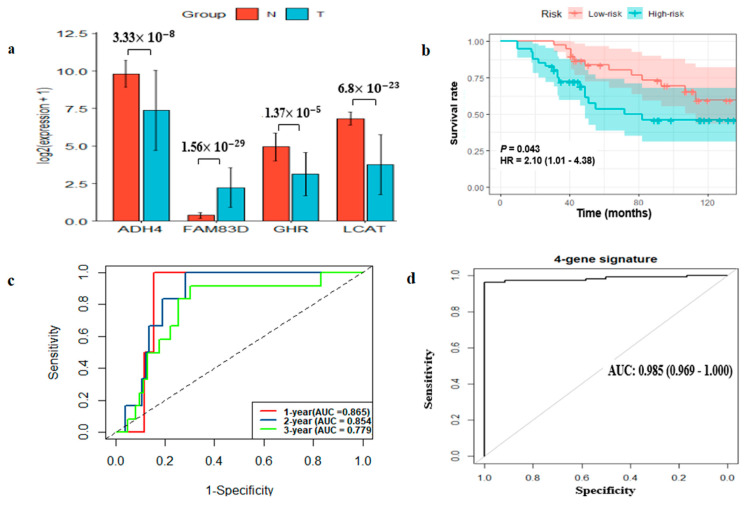
Comparison of expression level of the four-gene signature verified by our hospital samples. The relative expression level of *ADH4*, *LCAT*, *GHR* and *FAM83D* between HCC and normal tissues (**a**). Kaplan–Meier (KM) survival analysis for high-risk and low-risk subgroups classified by RS model (**b**). Verification of four-gene signature by time-dependent ROC analysis classified by RS model (**c**). The four-gene signature diagnostic performance in classifying HCC and normal tissue separated by RS model (**d**). RS, Risk score.

**Figure 7 ijms-22-01632-f007:**
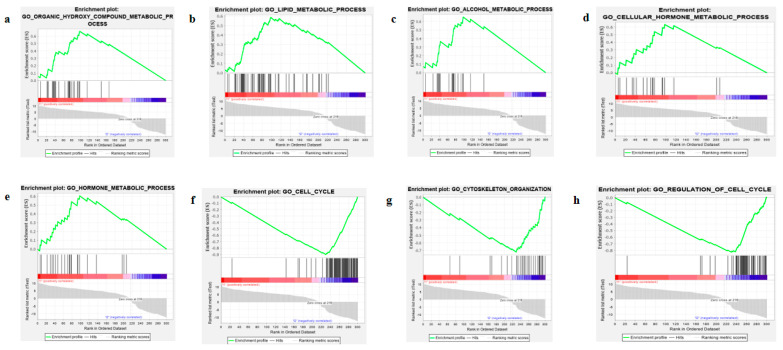
Gene set enrichment plot. GO-organic hydroxyl compound metabolic process (**a**), GO-lipid metabolic process (**b**), GO-alcohol metabolic process(**c**), GO-cellular and hormone metabolic process (**d**), GO-hormone metabolic process (**e**), GO-cell-cycle (**f**), GO-cytoskeleton organization (**g**), GO-regulation of cell cycle (**h**) profile of the running ES score and positions of GeneSet members on the rank ordered lists.

**Table 1 ijms-22-01632-t001:** The regression coefficients of survival-related genes identified by three penalized Cox methods.

Gene Symbol	Elastic Net	Lasso	Adaptive Lasso
*FAM83D*	0.091	0.104	0.113
*CDC20*	0.048	0.058	0.004
*TPX2*	0.016	-	0.084
*UBE2S*	-	-	0.024
*LECT2*	−0.001	-	-
*ANXA10*	−0.002	-	-
*DNASE1L3*	−0.004	-	-
*PON1*	−0.009	−0.008	-
*CD5L*	−0.009	−0.009	-
*CYP2C9*	−0.016	−0.016	-
*ADH4*	−0.019	−0.021	-
*CFHR3*	−0.021	−0.024	−0.008
*GHR*	−0.035	−0.037	-
*LCAT*	−0.080	−0.091	−0.135

**Table 2 ijms-22-01632-t002:** Statistical information of differential analysis of candidate genes (*GHR*, *ADH4*, *FAM83D* and *LCAT*). Log fold change (LFC) implies log fold change value and adjusted *p*-value less than 0.01.

Gene Symbol	Gene Full Name	GSE112790	GSE84402	GSE45267	TCGA
			LFC		
*GHR*	Growth Hormone Receptor	−2.371	−2.300	−2.632	−3.031
*LCAT*	lecithin-cholesterol acyltransferase	−3.359	−3.761	−2.935	−3.478
*FAM83D*	family with sequence similarity 83 member D	2.808	2.114	2.255	2.348
*ADH4*	Alcohol Dehydrogenase 4	−2.910	−2.328	−4.360	−4.360

**Table 3 ijms-22-01632-t003:** Univariate and multivariate Cox analyses of clinical pathological factors and risk model in TCGA and ICGC HCC datasets.

Factors	Univariate Analysis		Multivariate Analysis	
	HR	*p*-Value	HR	*p*-Value
**TCGA dataset**				
Age(>60 vs. ≤60)	1.27(0.88–1.85)	0.205		
Gender (Male vs. Female)	0.78(0.54–1.15)	0.209		
Stage (III+IV vs. I+II)	2.54(1.74–3.69)	<0.001	1.05(0.15–7.66)	0.960
T-stage (T3+T4 vs. T1+T2)	2.56(1.76–3.74)	<0.001	2.35(0.32–17.12	0.400
Grade (G3+G4 vs. G1+G2)	1.14(0.78–1.66)	0.506		
RS (High vs. Low risk)	2.28(1.54–3.36)	<0.001	2.20(1.48–3.25)	<0.001
**ICGC dataset**				
Age(>60 vs. ≤60)	1.28(0.70–2.35)	0.427		
Gender (Male vs. Female)	0.48(0.26–0.90)	0.022	0.38(0.20–0.73)	0.004
Stage (III+IV vs. I+II)	2.18(1.19–4.01)	0.012	2.66(1.42–5.01)	0.002
RS (High vs. Low risk)	3.52(1.77–6.99)	<0.001	3.57(1.78–7.18)	<0.001

Abbreviations: TCGA, The Cancer Genome Atlas; ICGC, International Cancer Genome Consortium, HR, hazard ratio; CI, confidence interval; RS, risk score.

**Table 4 ijms-22-01632-t004:** The results of significantly enrich KEGG pathways of prognostic genes and their co-expressed deregulated genes.

KEGG Terms	Count	FDR	Enriched Genes
hsa00830:Retinol metabolism	14	1.02 × $10^{-5}$	*CYP3A4, CYP2B6, CYP2C9, CYP2C8, ADH1C, CYP26A1, ADH1B, ADH1A, CYP1A2, RDH5, ADH4, AOX1, CYP2A6, RDH16*
hsa01100:Metabolic pathways	64	2.85 ×$10^{-5}$	*CYP3A4, NAMPT, CNDP1, OGDHL, ADH1C, ADH1B, ADH1A, ASPA, TDO2, MAT1A, ST3GAL6, GSTZ1, HPD, AADAT, ALDH6A1, FBP1, CYP26A1, CYP2E1, CYP1A2, ACADL, TAT, CTH, FOLH1, SDS, SQLE, RRM2, HAO2, BHMT, HSD11B1, CYP2A6, AKR1D1, SLC27A5, XDH, ASS1, CYP2B6, ALDOB, KMO, GLS2, TYMS, ACSL1, ADH4, DHODH, ENO3, DMGDH, ACSL4, CYP2C9, CYP2C8, NAT2, ACMSD, EPHX2, FTCD, IDO2, MAN1C1, PCK1, ACSM3, GBA3, AOX1, AGXT2, CYP4F3, CYP4F2, RDH16, CYP8B1, ACSM5, NNMT*
hsa04610:Complement and coagulation cascades	13	2.42 × $10^{-4}$	*F11, MBL2, C7, C9, MASP2, C6, F9, C1R, PLG, C8A, C8B, KLKB1, SERPINE1*
hsa05204:Chemical carcinogenesis	13	0.001	*CYP3A4, CYP2C9, CYP2C8, NAT2, ADH1C, ADH1B, ADH1A, CYP1A2, CYP2E1, CYP3A43, ADH4, HSD11B1, CYP2A6*
hsa00982:Drug metabolism—cytochrome P450	12	0.002	*CYP3A4, CYP2B6, CYP2C9, CYP2C8, ADH4, AOX1, ADH1C, ADH1B, CYP2A6, ADH1A, CYP2E1, CYP1A2*
hsa04110:Cell cycle	14	0.026	*CDC6, CDK1, TTK, CDC20, PTTG1, MCM2, MCM6, CCNB1, CCNE2, CCNB2,MAD2L1, BUB1, BUB1B, CCNA2*
hsa00980:Metabolism of xenobiotics by cytochrome P450	11	0.027	*CYP3A4, CYP2B6, CYP2C9, ADH4, HSD11B1, ADH1C, ADH1B, CYP2A6, ADH1A, CYP2E1, CYP1A2*
hsa00350:Tyrosine metabolism	8	0.036	*ADH4, AOX1, ADH1C, ADH1B, GSTZ1, ADH1A, TAT, HPD*

## Data Availability

All data used in this study can be downloaded from TCGA data repository (https://gdac.broadinstitute.org/), accessed on 1 January 2021. GEO data repository (https://www.ncbi.nlm.nih.gov/geo/), accessed on 1 January 2021 and ICGC repository (https://icgcportal.genomics.cn/), accessed on 1 January 2021. The data analyzed in this study were also partially is available from the corresponding authors on reasonable request.

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
