# Peer review of "Construction and Validation of a Prognostic Gene-Based Model for Overall Survival Prediction in Hepatocellular Carcinoma Using an Integrated Statistical and Bioinformatic Approach"

_ijms, 2021, doi:10.3390/ijms22041632_

Round 1

Reviewer 1 Report

In their manuscript "Construction and Validation of a Prognostic Gene-Based Model for Overall Survival Prediction in Hepatocellular Carcinoma Using an Integrated Statistical and Bioinformatic Approach" Dessie et al. present a study of constructing a novel prognostic gene-based model for predicting the prognosis of HCC patients using the TCGA dataset and took the ICGC dataset for validation. They found an AUROC of 0.992 for the TCGA data set.

General comments:

The topic is important, the methodology quite (but not breakingly) new.

The manuscript might need slight language editing by the journal staff.

For those unfamiliar with the data sets, a short description in the methods part is mandatory. E.g. it should be clearly stated, whether the genes were analyzed in normal tissues or tumors for all data sets, which stages were selected (if any), and what populations the data sets represent. How did you know about survival? What was the rationale to select just these data sets?

Also hints on the genetic analyses, on wich the data sets originate, might be useful here.

Specific comments (p =page, L=line)

  1. DEG short is never explained (differentially expressed genes?)
  2. p3L78: gnome
  3. p13L312: Syntax strange
  4. p14L321: Frist
  5. p14L321:How was the clinical data used, which data, ad for which purpose?
  6. p14L334: Multicollinearity?
  7. p15L351: All?
  8. p15L371: What is the mean survival time for HCC? Isn't 30d a bit short?
  9. p5L114: Awake?
  10. Figure 4: I would like to see confidence bands for the ROC curves.
  11. p8L185: The validation AUROC values are quite low - is this to be considered as a successful external verification?
  12. Figure 5c: What does the ROC curve show? cf. p13L286?  How does this compare to comment #11? To me, it is not the question here to distinguish HCC samples from normal tissue.

Reviewer 2 Report

This study is interesting, however a few changes are absolutely necessary.

1) The CMUH data must be made available via a publicly accessible link in the manuscript or needs to be uploaded to GEO and a link should be inserted in the manuscript. The authors themselves downloaded data from TGAC and ICGC to perform the study, and so they absolutely need to make the CMUH data available for download.

2) The authors must make all the scripts and commands that they used available as a supplementary file. This is necessary for reproducibility of the results.

Author Response

Reply to Reviewers’ Comments

Paper title: Construction and Validation of a Prognostic Gene-Based Model for Overall Survival Prediction in Hepatocellular Carcinoma Using an Integrated Statistical and Bioinformatic Approach

Authors: Eskezeia Y. Dessie, Siang-Jyun Tu, Hui-Shan Chiang, Jeffrey J.P. Tsai, Ya-Sian Chang, Jan-Gowth Chang and Ka-Lok Ng

Paper Number: ijms-1066301

Dear reviewers,

We would like to thank International Journal of Molecular Sciences for giving us the opportunity to revise our manuscript. We would also like to express our deepest appreciation for your time and efforts in reading our manuscript and providing helpful suggestions and comments. We have read the comments carefully and followed the suggestions in preparing the revised manuscript. The added/revised parts in the manuscript are highlighted in red color, and the replies are written in blue color for all reviewers. Our point-by-point responses to reviewers' comments are given below.

Thanks for all the help.

Best wishes,

Jan-Gowth Chang(Dr.)

Reviewer 2 Comments

Reply: We appreciate the reviewer's detail and careful reading of our manuscript. Our point-by-point responses to specific suggestions/ comments are in the following.

This study is interesting; however, a few changes are absolutely necessary. Concern #1) The CMUH data must be made available via a publicly accessible link in the manuscript or needs to be uploaded to GEO and a link should be inserted in the manuscript. The authors themselves downloaded data from TGAC and ICGC to perform the study, and so they absolutely need to make the CMUH data available for download.

Reply: We respect the reviewer’s appreciations. The reviewer's comment is also valuable. The main objective of this work is to develop prognostic and diagnostic model based on gene signature. We used three GEO datasets and TCGA dataset to identify differentially expressed genes, which helps to reduce noise arise from using different sequencing platforms and data selection. Then, based on the TCGA-HCC dataset, we developed a four-gene-signature prognostic model. Therefore, we only focused on using four-gene-signature to validate and verify our findings using ICGC and CMUH datasets. We did not perform genome-wide study of the CMUH and ICGC datasets. We can provide gene signature expression profiles and their corresponding clinical information (survival status) and time for public use.  The four-gene-signature expression profiles of HCC and normal tissue of samples  and the corresponding clinical information can be downloaded from https://github.com/estu2003/Prognostic-Gene-Based-Model.git or can be get from corresponding authors upon request.

Concern 2):  The authors must make all the scripts and commands that they used available as a supplementary file. This is necessary for reproducibility of the results.

Reply: We respect the reviews suggestion and comments to release the script command for reproducibility of the results. We deposited our scripts and commands in GitHub, which  can be assessed from https://github.com/estu2003/Prognostic-Gene-Based-Model.git.

Round 2

Reviewer 1 Report

The authors meticulously revised their manuscript and substantially improved it. I still have two comments:

  1. The title suggests the establishment of a prognostic model, and all the study design is oriented towards this topic. However, the authors also present a diagnostic model - personally, I consider this to be a bit misleading, as the superior AUROC value might be taken for the prognosis, and not for the discrimination (although it is stated): I actually do not see the added value of showing a diagnostic model: The question was prognosis, and the three AUROCS given for this sufficiently answer this question (with a fairly good predictive accuracy).
  2. The authors added confidence intervals, which I highly appreciate. However, these are not graphical confidence bands around the Kaplan Meier curves. I would advise the authors to add these (e.g. by bootstrapping techniques) to visualize if and where the curves are significantly different.

Author Response

Reply to Reviewers’ Comments

Paper title: Construction and Validation of a Prognostic Gene-Based Model for Overall Survival Prediction in Hepatocellular Carcinoma Using an Integrated Statistical and Bioinformatic Approach

Authors: Eskezeia Y. Dessie, Siang-Jyun Tu, Hui-Shan Chiang, Jeffrey J.P. Tsai, Ya-Sian Chang, Jan-Gowth Chang and Ka-Lok Ng

Paper Number: ijms-1066301

Dear reviewer,

We would like to thank International Journal of Molecular Sciences for giving us the opportunity to revise our manuscript. We would also like to express our deepest appreciation for your time and efforts in reading our manuscript and providing helpful suggestions and comments. We have read the comments carefully and followed the suggestions in preparing the revised manuscript. The added/revised parts in the manuscript are highlighted in red color, and the replies are written in blue color for all reviewers. Our point-by-point responses to reviewers' comments are given below.

Thanks for all the help.

Best wishes,

 Jan-Gowth Chang(Dr.)

Reviewer 1 Comments

Comments and Suggestions for Authors

The authors meticulously revised their manuscript and substantially improved it.

Reply: we appreciate the positive feedback.  Based the reviewer’s suggestion, we revised abstract[P1L31-L40], introduction [P2L61, L68-L69] and discussion[P13L304-L305] and conclusion[P18L443-448] in the revised manuscript. 

I still have two comments:

  1. The title suggests the establishment of a prognostic model, and all the study design is oriented towards this topic. However, the authors also present a diagnostic model - personally, I consider this to be a bit misleading, as the superior AUROC value might be taken for the prognosis, and not for the discrimination (although it is stated): I actually do not see the added value of showing a diagnostic model: The question was prognosis, and the three AUROCS given for this sufficiently answer this question (with a fairly good predictive accuracy).

Reply:  The main objective of this research is establishment of a prognostic model and based on prognostic model, we identified gene signature. Then, we validated the signature using training dataset, and external validation/verification dataset. Despite our main objective is prognosis prediction, it is also common to evaluate gene signature for their diagnostic performance. Scientific published works did this kind of validation to further examine the identified markers.  It will inform the reader that the role of the identified markers in diagnosis prediction ability and as the review suggested, we revised manuscript [in the abstract [P1L31-L40], introduction [P2L61, L68-L69] and discussion [P13L305-L306] and conclusion [P18L444-449] sections, focusing to our goal which is prognosis prediction]. We appreciate the valuable reviewer comments, which help us improve our manuscript.

  1. The authors added confidence intervals, which I highly appreciate. However, these are not graphical confidence bands around the Kaplan Meier curves. I would advise the authors to add these (e.g. by bootstrapping techniques) to visualize if and where the curves are significantly different.

Reply: Dear reviewer, we really appreciate your suggestions and your concern. However, overlay the AUROCS figure with multiple confidence intervals may not be accepted for publication standard quality figure, because multiple AUROCs confidence intervals cross each other. Moreover, based on the reviewer’s suggestion, we replotted the diagnostic performance of the AUROC curve for HCC from normal to see whether there is significance difference by bootstrapping technique using the CMUH dataset, and we found that there is no significance difference AUROC in estimating AUROCs using the bootstrapping technique (please check Figure 5 with confidence interval and Figure 5 without confidence interval in the revised manuscript). For the time-dependent AUROCs, we used “timeROC” package, there is no confidence band multiple as Kaplan Meier curves; whereas, Kaplan Meier curves was plotted with “Survival” and “survminer”, is possible graphical confidence bands around the Kaplan Meier curves. We checked previous scientific research and most published used similar AUCROCs were used to examine prognostic and diagnostic performance evaluation of predictive model based on the previous puplication {example:https://www.nature.com/articles/s41598-019-39273-4, https://journals.plos.org/plosone/article?id=10.1371/journal.pone.0232227, and https://www.mdpi.com/2077-0383/9/5/1540/htm and, Therefore, we understand your concern, while we left as it is due to the above reasons.        

Figure 5 with confidence interval                                          Figure 5 without confidence interval

Reviewer 2 Report

No further concerns

Author Response

Dear reviewer,

We really appreciate your feedback.

Thanks for all the help.

Best wishes,

Jan-Gowth Chang(Dr.)

This manuscript is a resubmission of an earlier submission. The following is a list of the peer review reports and author responses from that submission.

Round 1

Reviewer 1 Report

The publication is of a methodological nature. Paper has very high quality, and is well presented. I have no important substantive comments.

Reviewer 2 Report

This study cannot be evaluated due to the poor quality of english throughout the manuscript.